# The Inhibitory Role of miR-486-5p on CSC Phenotype Has Diagnostic and Prognostic Potential in Colorectal Cancer

**DOI:** 10.3390/cancers12113432

**Published:** 2020-11-19

**Authors:** Andrea Pisano, Carmen Griñan-Lison, Cristiano Farace, Giovanni Fiorito, Grazia Fenu, Gema Jiménez, Fabrizio Scognamillo, Jesùs Peña-Martin, Alessio Naccarati, Johannes Pröll, Sabine Atzmüller, Barbara Pardini, Federico Attene, Gabriele Ibba, Maria Giuliana Solinas, David Bernhard, Juan Antonio Marchal, Roberto Madeddu

**Affiliations:** 1Department of Biomedical Sciences, University of Sassari, 07100 Sassari, Italy; apisano@correo.ugr.es (A.P.); cfarace@uniss.it (C.F.); gfiorito@uniss.it (G.F.); g.fenu@studenti.uniss.it (G.F.); g.ibba@gmx.com (G.I.); gsolinas@uniss.it (M.G.S.); 2National Institute of Biostructures and Biosystems, 00136 Rome, Italy; 3Centre for Biomedical Research (CIBM), Biopathology and Regenerative Medicine Institute (IBIMER), University of Granada, 18100 Granada, Spain; carmengl@ugr.es (C.G.-L.); gemajg@ugr.es (G.J.); jespmar@ugr.es (J.P.-M.); 4Instituto de Investigación Biosanitaria Ibs.GRANADA, Organization University Hospitals of Granada, 18100 Granada, Spain; 5Excellence Research Unit Modeling Nature (MNat), University of Granada, 18016 Granada, Spain; 6MRC Centre for Environment and Health, Imperial College London, Norfolk Place, London W2 1PG, UK; 7Department of Human Anatomy and Embryology, Faculty of Medicine, University of Granada, 18016 Granada, Spain; 8O.U. of Surgery I (Surgical Pathology), A.O.U. Sassari, 07100 Sassari, Italy; fscognamillo@uniss.it (F.S.); f.attene@uniss.it (F.A.); 9Molecular Epidemiology and Exposome Research Unit, Italian Institute for Genomic Medicine (IIGM), c/o IRCCS Candiolo, Candiolo, 10060 Torino, Italy; alessio.naccarati@iigm.it (A.N.); barbara.pardini@iigm.it (B.P.); 10Molecular Epidemiology and Exposome Research Unit Candiolo Cancer Institute, FPO-IRCCS, Candiolo, 10060 Torino, Italy; 11Austrian Cluster for Tissue Regeneration, 1200 Vienna, Austria; johannes.proell@jku.at; 12Center for Medical Research, Johannes Kepler University, 4040 Linz, Austria; sabine.atzmueller@jku.at; 13Red Cross Blood Transfusion Service, 4020 Linz, Austria; 14Division of Pathophysiology, Institute of Physiology and Pathophysiology, Medical Faculty, Johannes Kepler University, 4040 Linz, Austria; david.bernhard@jku.at

**Keywords:** miR-486-5p, colorectal cancer, metastasis, cancer stem cells, serum and stool biomarkers

## Abstract

**Simple Summary:**

Colorectal cancer is nowadays one of the most diffuse cancers worldwide. Its early diagnosis is often linked with a positive outcome. However, colorectal cancer is a “silent” cancer and the first symptoms usually appear when this is already in an advanced stage. To this end, an improvement in diagnostic, prognostic methods is necessary. This scenario fits the present study, which through a multidisciplinary approach, involving the study of samples of human origins and in vitro studies, seeks to clarify the involvement of microRNAs in the regulation of cancer stem cells of colorectal cancer. The oncosuppressive role of miR-486-5p is here extensively investigated, supporting the hypothesis of its role in the biology of cancer stem cells, giving a contribution for further studies to investigate its possible use as a diagnostic and prognostic biomarker of colorectal cancer.

**Abstract:**

Colorectal cancer (CRC) is the third most frequent cancer worldwide and the second cause of cancer deaths. Increasing evidences supports the idea that the poor prognosis of patients is related to the presence of cancer stem cells (CSCs), a cell population able to drive cancer recurrence and metastasis. The deregulation of microRNAs (miRNAs) plays a role in the formation of CSC. We investigated the role of hsa-miR-486-5p (miR-486-5p) in CRC, CSCs, and metastasis, in order to reach a better understanding of the biomolecular and epigenetic mechanisms mir-486-5p-related. The expression of miR-486-5p was investigated in three different matrices from CRC patients and controls and in CSCs obtained from the CRC cell lines HCT-116, HT-29, and T-84. In the human study, miR-486-5p was up-regulated in serum and stool of CRC patients in comparison with healthy controls but down-regulated in tumor tissue when compared with normal mucosa. miR-486-5p was also down-regulated in the sera of metastatic patients. In vitro, miR-486-5p was down-regulated in CSC models and it induced an inhibitory effect on stem factors and oncogenes in the main pathways of CSCs. Our results provide a step forward in understanding the role of mir-486-5p in CRC and CSC, and suggest that further studies are needed to investigate its diagnostic and prognostic power, possibly in combination with other biomarkers.

## 1. Introduction

Colorectal cancer (CRC) is a malignancy of the gastrointestinal tract. More than 90% of colorectal carcinomas are adenocarcinomas, originating from the epithelial cells of the colorectal mucosa [1]. CRC is the third most frequent cancer worldwide, with 1.85 million new cases a year, and the second most deadly, with 881,000 deaths estimated in 2018 [2]. More than two-thirds of all cases and about 60% of all deaths from CRC occur in countries with a high or very high human development index. However, in recent years, this trend is reversing. In countries with high human development indices, such as the USA, Japan, Austria, and France, both the mortality rate and incidence of CRC are progressively decreasing in response to improvements in diagnostic procedures, whereas in countries with low development indices, such as the Philippines, China, Brazil, and others, the mortality rate and incidence of CRC are increasing in response to the Westernization of these areas [3]. The worst prognosis in CRC patients is associated with molecular mechanisms related to the propagation of cancer stem cells (CSCs). According to the CSC model, proposed for the first time 150 years ago [4] and first supported by evidence that emerged in 1994 [5], tumors are initiated and propagate by the clonal division of mutated stem cells, which acquire tumor behavior [6]. Other evidence suggests that CSCs can be generated by differentiated cells that acquire stem behavior by activating stem-related molecular pathways and processes e.g., the epithelial to mesenchymal transition (EMT) [7]. The CSC phenotype is present in a small subpopulation of cells in CRC, predominantly occurring at the bottom of the colonic crypts [6,8]. CSCs play crucial roles in metastasis, both lymphatic and distant, tumor recurrence, and drug resistance through several biomolecular mechanisms, including self-renewal and differentiation, multipotency, cell quiescence, angiogenic induction, EMT, and immune evasion [9,10]. An early-stage diagnosis is essential for a positive prognosis in CRC [11]. Colonoscopy and the histological analysis of tumor biopsies are currently considered the gold standard methods for diagnosis [12]. Various techniques are used in CRC screening, including faecal immunochemical testing, high-sensitivity guaiac-based faecal occult blood testing, multi-target stool DNA testing, colonoscopy, computed tomographic colonography, and flexible sigmoidoscopy [13]. Prognostic techniques include TNM classification, carcinoembryonic antigen (CEA) quantification, evaluation of microsatellite instability, and mutations in *KRAS* (KRAS proto-oncogene, GTPase), *NRAS* (NRAS proto-oncogene, GTPase), *BRAF* (v-RAF murine sarcoma viral oncogene homolog B), and mismatch repair (MMR) genes [12]. However, the available diagnostic and prognostic methods have some limitations. In some cases, these procedures are invasive and require the tolerance and adequate patient preparation, as in the case of colonoscopy [14]; or they are moderate sensitive, as in the case of screening tests [12]. Therefore, improvements in these fields are necessary. Recent evidence indicates that microRNAs (miRNAs) are good candidate markers for diagnostic and prognostic procedures [15,16]. miRNAs are short (18–22 nt) single-stranded RNAs that bind to target mRNAs and inhibit their translation. Altered miRNA expression levels have been shown to drive a plethora of diseases, including cardiovascular diseases [17], diabetes [18], and cancer [19] in which they can act both as oncogenes [20] and oncosuppressors [21]. The altered expression of miR-486-5p has been observed in different tumors [22,23]. miR-486-5p is described as an oncosuppressor in CRC because it is progressively down-regulated in tissues as the pathology progresses [24], whereas the opposite behavior is described in blood [24,25]. The overexpression of miR-486-5p in blood has been proposed as a diagnostic biomarker of CRC [24]. However, the effects of miR-486-5p in CRC and in the biology of CSC, as well as its role as a prognostic factor in blood or other biological matrices in CRC patients, remain poorly understood. In this study, we first characterised the miRNome of control subjects and CRC patients and evaluated the expression levels of miR-486-5p in the sera and stool samples from healthy individuals and CRC patients. We also conducted a meta-analysis of sera and tissues data from online datasets at the Gene Expression Omnibus (GEO) database [26]. Finally, we investigated the role of miR-486-5p in vitro in CSC models generated from the colorectal cell lines HCT-116, HT-29, and T-84. The expression of miR-486-5p was analysed in CSC subpopulations and monolayer cultures, and its role in the induction of CSC stemness properties was established with the use of miR-486-5p mimic or inhibitor, to increase its regulatory effects or inhibit its action, respectively. All the steps are resumed in the following workflow (Figure 1).

## 2. Results

### 2.1. Serum Small RNA-Sequencing

The sequences from the pooled serum RNA samples of CRC patients showed differential expression levels of miR-486-5p, with a progressive reduction in its expression as the disease progressed. The highest expression levels were observed in the non-tumor group (average number of copies 1,364,721, standard error 57,572), and the lowest in the metastatic group (average number of copies 752,483, standard error 61,730) which includes patients with both lymphatic and distant metastasis. We observed statistically significant differences between the metastatic and non-tumor groups (log_2_ FC = −0.85, *p* = 0.0008; FDR-adjusted *p* = 0.03), the tumor and non-tumor groups (log_2_ FC = −0.63, *p* = 0.007, FDR-adjusted *p* > 0.05), and the metastatic and tumor groups (log_2_ FC = −0.22, *p* = 0.03; FDR-adjusted *p* > 0.05). The *p*-value was 0.008 in ANOVA test for the comparison among the three groups. The expression of hsa-miR-342-3p, used as the reference miRNA for normalisation of miR-486-5p, was similar in all the three groups: non-tumor (average number of copies 3069, standard error 147), tumor (average number of copies 3074, standard error 24), and metastatic (average number of copies 3074, standard error 574) (Figure 2). Table 1 shows the data of the most significantly dysregulated miRNAs.

### 2.2. MiR-486-5p Expression in Sera Determined with Real-Time RT–PCR

Using the average expression of hsa-miR-342-3p, hsa-223-3p, and hsa-miR-93-5p as reference for normalisation, we confirmed the trend previously reported in the tumor vs. non-tumor group comparison (FC = 2.2, *p* > 0.05) and in the metastatic vs. non-tumor group comparison (FC = 0.89, *p* > 0.05). Furthermore, we confirmed the statistically significant differences observed between tumor and metastatic groups (FC = 2.47, *p* = 0.02) (Figure 3a). Using hsa-miR-342-3p as reference miRNA for normalization, we observed a differential expression of miR-486-5p in the tumor (FC = 2.73, *p* > 0.05) and metastatic CRC groups (FC = 0.79, *p* > 0.05) with respect to controls. There was also a statistically significant difference between tumor and metastatic groups (FC = 3.45; *p* = 0.02; Appendix A). The control group was also compared with the total tumor group (stages I–IV): we observed an up-regulation of miR-486-5p in CRC patients when for normalisation was used the average of hsa-miR-342-3p, hsa-223-3p, and hsa-miR-93-5p (Figure 3b, FC = 1.35), or the hsa-miR-342-3p reference alone (Appendix A, FC = 1.47).

### 2.3. Mir-486-5p Levels in Stool by Small RNA-Sequencing

We found that miR-486-5p was overexpressed in the stools of the CRC patients compared with controls. The average number of reads were 43.65 in the controls and 223.67 in the CRC patients, with log_2_ FC = 2.36 and adjusted *p* = 3.6 × 10^−11^ (Figure 3c).

### 2.4. Meta-Analysis

We performed a meta-analysis of the expression levels of miR-486-5p in serum and tissue samples from CRC patients and healthy controls using datasets from the GEO database. In serum, standardized difference between cases and controls of 1.01 (95% confidence interval 0.70; 1.32) was observed for miR-486-5p (Figure 4a). In tissue samples, a standardized difference between cases and controls risk ratio of −0.61 (95% confidence interval −0.70; −0.52) was observed for miR-486-5p (Figure 4b). These data indicate that miR-486-5p was overexpressed in the sera of patients relative to the controls, whereas it was down-regulated in tumor tissues in comparison with healthy tissues. In dataset GSE39845, a difference in miR-486-5p expression was observed between early-stage (stage I–II) and late-stage CRC (stage III–IV), and it was progressively down-regulated with disease progression (Figure 4c), although the difference was not statistically significant.

### 2.5. miR-486-5p Expression in Cell Lines by Real-Time RT–PCR Analysis

The expression of miR-486-5p was evaluated in an in vitro model able to distinguish its relationship with cancer, in both stem and non-stem cells. The Ct values for miR-486-5p in the CSC and monolayer models of HCT-116, HT-29 and T-84 cell lines were obtained. Then, the FC values between CSCs and the respective monolayers for each cell line were calculated. The levels of miR-486-5p were markedly lower in the three CSC models than in the corresponding monolayers (Figure 5a).

### 2.6. Effect of miR-486-5p Mimic and Inhibitor on Expression of Stemness Genes

The expression of five stemness genes, *SOX2*, *NANOG*, *OCT4*, *KLF4*, and c*MYC*, was evaluated in HCT-116 CSCs and monolayer cells, under the effect of either the miR-486-5p mimic or inhibitor and normalised to the respective controls. We focused our attention on the HCT-116 cell line because, it has been shown as a better model to induce the CSC phenotype, compared to HT-29 which have a greater tendency to differentiate more fast and precisely in vitro than HCT-116 [27], whereas T-84 cell line is a CRC in vitro model derived from lung metastasis. In the HCT-116 monolayers, the expression levels of *OCT4* and *KLF4* were higher in the controls than in the mimic-treated cells, whereas the expression of all the genes was higher in the inhibitor-treated cells than in either the controls or the mimic-treated cells with exception of *SOX2*, which was slightly more expressed in the mimic-treated cells, although with not statistical significance. The expression levels of *OCT4*, *KLF4*, and c*MYC* were significantly different between cells in all treatment comparisons (Figure 5b and Appendix A). In the HCT-116 CSCs, the expression of all the stem factors was higher in all the controls than in the mimic-treated cells, with the exception of *SOX2*, whereas the expression of all the genes was higher in the inhibitor-treated cells than in the control and mimic-treated cells. The differences in all the comparisons were statistically significant, except for the difference in *OCT4* among the control and mimic-treated cells (Figure 5c and Appendix A). Comparing the effect of control mimic on stem cell genes in monolayers and CSCs, it shows overexpression of stem cell genes in CSCs, as expected (Figure 5d); the same effect was evaluated for the counter inhibitor, in this case an upregulation is observed for most of the factors except for Nanog and Oct4 (Figure 5e).

### 2.7. Microarray Genomic Analysis of the miR-486-5p Effect

From our results, we conceived a working hypothesis: miR-486-5p exerts a negative regulatory effect on the CSC phenotype in CRC through the activation of genes that inhibit the CSC phenotype and through the down-regulation of genes that promote the CSC phenotype. To test this working hypothesis (depicted in Figure 6), we used mRNA microarrays to analyse the effects of the induced up- or down-regulation of miR-486-5p on HCT-116 cells, in both monolayer cells and CSCs (described in Section 4.10). This approach allowed us to observe if some of these genes suffered the same effect in the two cell types and if they were progressively inhibited or promoted in the passage from the monolayer and CSC phenotype. We tested 135,750 transcripts, among which 2575 showed expression patterns consistent with our working hypothesis (empirical *p* < 0.05 after permutation-based correction for multiple testing). These genes were then filtered to a list of 240 genes with crucial roles in the CSC phenotype, derived from the four principal biological pathways involved in CSCs: Wnt, Notch, Hedgehog, and TGF-β. Among these genes, 16 were consistent with our working hypothesis. The weighted Kolmogorov–Smirnov enrichment test indicated that genes adherent to our working hypothesis are enriched in the four biological pathways previously described (empirical *p* < 0.01). These results indicate that miR-486-5p regulate the expression of important genes pertaining to CSC-related pathways. miR-486-5p promotes the expression of 11 genes (PLK1, FRZB, TCF7L2, SMAD4, NFATC1, KDM4C, PPARD, KLF4, CDX2, STAT3, and EP300), which were up-regulated when miR-486-5p was overexpressed and down-regulated when it was inhibited. On the other hand, miR-486-5p inhibits the expression of 5 genes (NRCAM, JUN, WNT10A, FZD8, and NODAL) which were down-regulated when miR-486-5p was overexpressed and up-regulated when it was inhibited. All these genes showed the same behavior in monolayer cells and CSCs, when analysed independently. Eight of these genes showed progressive up or down-regulation in the transition from monolayer cells to CSCs with p_trend_ < 0.05: PLK1 (p_trend_ = 0.0001), FRZB (p_trend_ = 0.0002), TCF7L2 (p_trend_ = 0.0004), SMAD4 (p_trend_ = 0.016), NFATC1 (p_trend_ = 0.02), KDM4C (p_trend_ = 0.04), NRCAM (p_trend_ = 0.008), and JUN (p_trend_ = 0.04). The other eight genes showed no progressive up- or down-regulation in the transition from the monolayer to the CSC phenotype, then the *p* values were calculated for each condition: PPARD (monolayer p_trend_ = 0.09; CSCs p_trend_ = 0.01), KLF4 (monolayer p_trend_ = 0.09; CSCs p_trend_ = 0.03), CDX2 (monolayer p_trend_ = 0.02; CSCs p_trend_ = 0.06), STAT3 (monolayer p_trend_ = 0.08; CSC p_trend_ = 0.01), EP300 (monolayer p_trend_ = 0.09; CSCs p_trend_ = 0.03), WNT10A (monolayer p_trend_ = 0.06; CSCs p_trend_ = 0.02), FZD8 (monolayer p_trend_ = 0.09; CSCs p_trend_ = 0.05), and NODAL (monolayer p_trend_ = 0.01; CSCs p_trend_ = 0.03) (Figure 7).

## 3. Discussion

This study aimed to improve our knowledge about the biomolecular and epigenetic mechanisms regulated by miR-486-5p in CRC and CSCs, and to possibly, provide useful information for further studies in the field of biomarker discovery. To investigate whether differential miRNA expression occurs in CRC, we performed a preliminary analysis investigating the whole miRNome in serum samples of 47 subjects including healthy controls, patients with low-stage, non-metastatic CRC (stage I–II), and with advanced stage, metastatic CRC (stage III–IV, referring both to lymphatic and distant metastasis). From this analysis, we detected an interesting profile of miR-486-5p, a miRNA scarcely studied in CRC, prompting its further investigation. miR-486-5p was progressively down-regulated in sera as the CRC pathology progressed, with the lowest expression levels in the metastatic group. To confirm these data, we quantified miR-486-5p in individual patients by real-time RT–PCR, from which two main results emerged: first, the real-time RT-PCR analysis overturned the difference between the controls and the non-metastatic tumor group observed when miRNAs were profiled by small RNA-sequencing. These results were in contrast, and this is probably attributable to the different analysis approach used in the two techniques: contrary to real-time RT-PCR, in miR–seq the samples were merged in pool per group, and the presence of outlier patients may have induced bias. Furthermore, this data was not supported by statistical significance. More specifically, miR-486-5p was down-regulated in the control group in relation to the non-metastatic tumor group (stage I–II). When the CRC groups were merged in a single group (all stages) and compared with the control group, miR-486-5p was up-regulated in the tumor group. An up-regulation of miR-486-5p in the sera of CRC patients was strongly supported by a meta-analysis of data on thousands of patients. In order to test the diagnostic power of miR-486-5p we performed a ROC miR-486-5p in the sera of our patients (Appendix A), in a model which includes other predictor factors such as age, sex, smoking history, levels of CEA, and levels of alkaline phosphatase. Further, we investigated the prognostic utility of mir486-5p measured in sera of our CRC patients via a Cox regression model to evaluate 5-years survival after surgery (Appendix A). The model included age, sex, tumor stage and grade, CEA, and alkaline phosphatase as covariates. Our results (details about methods and results are indicated in the Appendix A) indicate that the inclusion of miR-486-5p in a prediction model moderately increased both prognostic and diagnostic power (although without strong statistical significance, likely as a consequence of the lack of statistical power). Although we are aware of the limitations of these statistical analyses performed on small sample size (being not the primary aim of this study), our results suggest miR486-5p deserves further investigation to understand whether it can become a useful diagnostic and prognostic biomarker, possibly in combination with those currently used in clinical practice. To better understand the behavior of miR-486-5p in CRC, we compared its expression in stool samples from CRC patients and controls, to establish its usefulness in CRC screening [12,13], and in the intestinal mucosa tissues affected by the disease. The stool samples were analysed with small RNA-sequencing in another cohort of CRC patients and controls. The results were consistent with those observed in serum, showing an up-regulation of miR-486-5p in CRC patients. However, in tissue samples, the expression levels of miR-486-5p were in the opposite direction compared to what was observed in serum samples. The second interesting result was that our real-time RT–PCR analyses validated the small RNA-sequencing results, confirming the lower expression of miR-486-5p in the sera of patients with metastatic CRC than in those with low-stage CRC. Our findings have been scarcely investigated in the literature and can represent an incentive to further investigate its utility as prognostic factor. In consideration of the important role of the CSC in metastasis of CRC and other tumors [9,10], we decided to investigate the role of miR-486-5p in this population. To this end, CSC models were established from three CRC cell lines HCT-116, HT-29, and T-84. Real-time RT–PCR confirmed that the expression of miR-486-5p was down-regulated in these models relative to that in monolayer cells. The inhibitory role of miR-486-5p in CSCs was further supported by a functional study of HCT-116 cells, in which the up- and down-regulation of miR-486-5p was induced in both the cellular models. The effects of miR-486-5p on stemness factors: Sox2, Oct4, Klf4, cMyc and Nanog were then examined by real-time RT–PCR, and the effects on the four principal pathways of CSCs (Wnt, Hedgehog, Notch, and TGF-β) were analysed by microarray analysis.

The up-regulation of miR-486-5p in blood has been previously observed in cancer patients, including those with lung cancer [28] and CRC [24,25]. We observed this trend in a group of patients and controls and validated it with an extensive meta-analysis. Importantly, our hypothesis was also confirmed in stool samples. The presence of occult blood in faecal samples is routinely used in CRC screening [12]; furthermore, correlations between the concentrations of other miRNAs in blood and stool have been demonstrated, as for miR-486-5p [29]. However, an opposite result was observed in tissues as the expression of miR-486-5p was higher in healthy mucosae than in tumor. The lower expression of miR-486-5p in CRC tissues has been previously reported in the literature, where this miRNA was described as an oncosuppressor [22,23,24]. Liu et al. recently discussed the contrasting behavior of miR-486-5p in blood components and tissues [24]. Many other authors have investigated the opposite behaviors of certain miRNA expression levels in tissues and serum [30,31]. In some cases, this phenomenon is attributable to an endocrine role played by miRNAs released from the primary tumor into the bloodstream by exosomes, conditioning other cells, or the tumor microenvironment [31,32]. In contrast, some miRNAs are released from cells in a non-exosomal way and are carried by proteins, such as AGO [33,34] and apolipoproteins [35], via active or passive means, such as leakage from cells after injury, chronic inflammation, apoptosis, or necrosis [34,36]. This mechanism is considered to account for 90% of circulating miRNAs [37]. Turchinovic et al. assumed that the greatest amounts of extracellular miRNAs do not occur in exosomes but are conjugated to AGO proteins, and are released into the bloodstream after cell death from necrosis or apoptosis, and they attributed a role in these phenomena to some of the released miRNAs [33]. These observations may clarify why miR-486-5p is down-regulated in the sera of patients with metastatic CRC. We hypothesized a link between the down-regulation of miR-486-5p in serum and its absence in CSCs, as observed in our study. It is conceivable that the high serum levels of miR-486-5p in the early stages of CRC patients could be associated with its release after cell death [33], which may be induced by the miRNA itself during the performance of its tumor-suppressive function [22,23], such as in apoptosis [38]. Alternatively, the absence of miR-486-5p can drive to a worsening of cell fate, a condition that may have a potential role in CSC development and metastasis insurgence, although more specific and in-depth studies are needed to demonstrate this hypothesis.

The role of miR-486-5p in the CSC phenotype and in metastasis has been described in tissues from different tumors [22,23], although in glioblastoma, the opposite behavior has been described [39]. In the present study, we also evaluated the effects of miR-486-5p on the biology of colorectal CSCs. The modulation of expression of *SOX2*, *OCT4*, c*MYC*, *KLF4* [40], and *NANOG* [41] by treatment with an miR-486-5p mimic or inhibitor suggests that miR-486-5p inhibits the expression of the main genes involved in stemness. miR-486-5p predominantly negatively affected the expression of *KLF4* and *OCT4* in both monolayer cells and CSCs, followed by c*MYC* and *NANOG*, for which the greatest effect was observed in CSCs; however, no clear effect on *SOX2* expression was observed. KLF4 is involved in a series of CSC processes, including cell differentiation, the maintenance of pluripotency, self-renewal, and apoptosis [42,43], and it is overexpressed in CSCs of CRC [43] so that several studies have described it as an oncosuppressor in CRC [44,45]. Furthermore, KLF4 promotes a series of key metastatic mechanisms, such as cell migration and invasion, the EMT [42], and the inhibition of TP53, with the consequent promotion of *NANOG* expression [40,45]. The inhibitory effect of miR-486-5p was also observed on *OCT4* expression. This effect has already been observed in liver cancer [22] and highlights the involvement of miR-486-5p in CSC inhibition. In fact, *OCT4* is overexpressed in cancer [46], including CRC, where it is involved in malignancy and metastasis [47]. Moreover, OCT4 in combination with SOX2 regulates NANOG in the stemness process [40,41,42]. *NANOG* is a dispensable factor in the induction of stemness and is also important in its maintenance [40]. This evidence indicates a late activation of *NANOG* in the stemness process in accordance with our results, in which the effects of miR-486-5p on NANOG were more evident in CSCs. Other authors have demonstrated the role of *NANOG* in CSCs and CRC [48]. Like *NANOG*, in the present study, the most evident effect of c*MYC* was observed in CSCs. c*MYC* regulates nearly 15% of all human genes [49] and is involved in the biology of CSCs [50]. The effect of miR-486-5p on c*MYC* has been previously observed in CRC [51]; furthermore, miRpath indicates c*MYC* as a direct target of miR-486-5p [52]. As mentioned before, we did not detect a clear effect of miR-486-5p on *SOX2* expression. SOX2 is a transcription factor involved in embryonal development, the maintenance of pluripotency, and cell self-renewal [40], with roles both in tumors [53,54], including CRC [47,55], and in CSC development [42]. The relationship between *SOX2* and miR-486-5p is controversial and, although *SOX2* expression is promoted by miR-486-5p in some tumors, such as glioblastoma [39], it inhibits *SOX2* in liver cancer [22]. This corroborates our observation that there is no link between miR-486-5p and *SOX2* expression.

The CSCs phenotype is regulated by four principal pathways: Wnt, Notch, Hedgehog, and TGF-β [6]. Our microarray analyses data highlighted the involvement of miR-486-5p in CRC biology and in the molecular mechanisms of CSCs by regulating the above-mentioned pathways. miR-486-5p promotes the expression of five genes with oncosuppressive role and thereby with a negative effect on the CSC phenotype: *FZB* [56,57], *SMAD4* [58,59], *NFATC1* [60,61], *CDX2* [62,63] and *EP300* [64,65]. On the other hand it inhibits the expression of further five genes with oncogenic and CSC-promoting roles: *NRCAM* [66,67], *JUN* [68,69], *WNT10A* [70,71], *FZD8* [72,73], and *NODAL* [74,75]. Furthermore, miR-486-5p showed a positive effect on the expression of PLK1, involved with a proper cell division process [76] and with controversial behavior in cancer, in which it is described both as an oncogene [77] and an oncosuppressor [78]. Finally, miR-486-5p also promote the expression of five genes with promoting cancer roles: *KDM4C* [79,80], *PPARD* [81,82], *KLF4* [43], *STAT3* [22,40], and *TCF7l2* [83,84]. The up-regulation of *STAT3* and *TCF7l2* in the presence of this miRNA may be a consequence of the miR-486-5p-mediated activation of *EP300* [63] and *FZB* [85] (Figure 8).

## 4. Materials and Methods

### 4.1. Patient Information

miR-486-5p expression levels were investigated in clarified sera and stool from CRC patients in a retrospective study. The serum samples used in the analysis originated from 47 individuals (University of Sassari, Sassari, Italy) divided into three groups: controls (non-tumor n = 10; average age = 64.6 years; 60% male, 40% female), CRC (patients with stage I–II colorectal adenocarcinoma without metastasis, n = 13; average age = 70.6 years; 46.2% male, 53.8% female), and metastatic CRC (patients with stage III–IV colorectal adenocarcinoma with both lymphatic and distant metastasis, n = 24; average age = 69.6 years; 54.2% male, 45.8% female). The stage merging was done to distinguish patients in which cell migration and invasion already occurred from those in which it did not. The TNM 0–IV classification was used to establish cancer staging with histological methods [12].

In parallel, we performed small RNA-sequencing to annalys the whole miRNome in 137 stools samples from 79 healthy subjects (mean age = 57.4 years; 50.7% male, 49.3% female) and 58 CRC patients (mean age = 71 years; 70.7% male, 29.3% female). Samples were collected from patients recruited in a hospital-based study at the Clinica S. Rita in Vercelli, Italy. Naturally, evacuated stool samples were obtained from all patients after they were instructed to self-collect the specimens at home before any bowel preparation for colonoscopy. Stool were collected in the Stool Nucleic Acid Collection and Preservation Tubes (Norgen Biotek Corp., Thorold, ON, Canada) with RNA stabilising solution, and returned to the hospital the day of the colonoscopy or at the time of blood sampling. Aliquots (200 mL) of the stool samples were stored at −80 °C until RNA and DNA extraction. The study was approved by the local Ethics Committees (University of Sassari V. le S. Pietro 43/C, 07100 Sassari, Italy and Azienda Ospedaliera SS, Antonio e Biagio e C. Arrigo of Alessandria, Italy), and informed consent was obtained from all the participants.

### 4.2. RNA Extraction from Patient Sera and Stool Samples

Before surgery or any treatment, 5 mL of blood were collected from each patient and placed in an S-Monovette^®^ 9 mL, Serum Gel with Clotting Activator tube (Sarstedt, Nümbrecht, Germany). RNA was extracted from the serum samples using the miRNeasy^®^ Serum/Plasma Kit (Qiagen, Hilden, Germany), according to the manufacturer’s instructions, as previously described [19]. RNA was extracted from the stool samples with a Stool Total RNA Purification Kit (Norgen Biotek Corp.), as described by Tarallo et al. [86].

### 4.3. Small RNA-Sequencing in Serum

To analyse the expression levels of the whole miRNome in different tumor phases, we pooled the RNAs extracted from the sera from the three groups: non-tumor (n = 10), tumor (n = 13), and metastatic (n = 24). The final sample from each group had a final volume of 11 µL and a yield of 1 µg of RNA, and each initial sample contributed with an equal volume to the final pool. The pools were analysed in duplicate in service at the Centre for Genomics and Oncological Research (GENYO; Pfizer–University of Granada–Andalusian Regional Government, Granada, Spain). Standard quality control steps were included to determine the quantity and quality of the total RNA with a NanoDrop™ 2000 spectrophotometer (Thermo Fisher Scientific, Waltham, MA, USA) and electrophoresis on an Agilent 2100 Bioanalyzer, with the Eukaryote Total RNA Pico Kit (Agilent Technologies, Santa Clara, CA, USA), and a High Sensitivity DNA Assay (Agilent Technologies). Libraries were prepared from 50 ng of extracted RNA with the NEXTflex™ Small RNA-Seq Kit v2 (Bioo Scientific Corp., Austin, TX, USA). To prepare small-RNA libraries, we simultaneously multiplexed 12 samples using 22 cycles of amplification, and the miRNAs were selected by size (130–170 bp; or 16–56 bp without adaptors) in an acrylamide gel. All the libraries were sequenced with the NextSeq 550 System (Illumina, San Diego, CA, USA) using a High Output Kit v2 (Illumina) with 75 cycles. All samples were read in a single run.

### 4.4. Real-Time RT-PCR Assay for miRNA Expression Profiling in Sera

miRNA expression levels are generally at very low concentrations in serum. Hence, to minimise any technical bias, we used Custom PCR Panels (Qiagen) with pre-spotted selected primers for the real-time reverse transcription (RT)–PCR-based miRNA expression analysis. Serum RNA samples from each patient were diluted in nuclease-free water to a final concentration of 5 ng/µL and cDNA was synthesised with the miRCURY™ LNA™ RT Kit (Qiagen), according to the manufacturer’s instructions. The reactions were spiked in with exogenous UniSp6 RNA (RNA Spike-In Kit; Qiagen). The RT protocol consisted of 60 min at 42 °C, 5 min at 95 °C, and cooling to 4 °C. The cDNA was immediately stored at −20 °C until processing. miRCURY LNA miRNA Custom PCR Panels (Qiagen) were furnished pre-spotted with the primers hsa-miR-223-3p, hsa-miR-93-5p, miR-486-5p, and hsa-miR-342-3p in 96-well plates. In the real-time RT-PCR assays, the cDNA was diluted 1:40 in nuclease-free water and 4 µL of diluted cDNA was mixed with 5 µL of PCR Master Mix and 1 µL of each primer. The thermal cycling conditions included a melting curve analysis and were run in the StepOne Real-Time PCR System (Thermo Fisher, Waltham, MA, Italy) with the following parameters: 95 °C for 10 min, followed by 45 cycles at 95 °C for 10 s and 60 °C for 1 min, with a ramp rate of 1.6 °C/s. Human miR-342-3p (hsa-miR-342-3p) was used as the reference miRNA, as previously described [87], and because its expression was constant in all three pools analysed with small RNA sequencing (miR-seq). The expression levels were analysed also using the average expression levels of hsa-miR-342-3p, hsa-miR-223-3p, and hsa-miR-93-5p as references, according to the method described by Mestdagh et al. [88]. Both the normalization methods -with hsa-miR-342-3p and the average expression levels of hsa-miR-342-3p, hsa-miR-223-3p, and hsa-miR-93-5p- were taken in consideration.

### 4.5. Small RNA-Sequencing in Stool

Libraries were prepared from the total RNA extracted from the stool samples with a NEBNext^®^ Multiplex Small RNA Library Prep Set for Illumina (protocol E7330; New England BioLabs Inc., Ipswitch, MA, USA), as previously described [86,89].

### 4.6. Meta-Analysis of miR486-5p Serum and Tissues from public Database

Ten datasets of miRNA expression levels in human CRC patients versus controls (five for serum and five for tissues) were selected from the GEO database [26]. The serum datasets were: gse59856 (151 healthy individuals and 50 CRC patients), gse106817 (2724 healthy individuals and 150 CRC patients), gse112264 (41 healthy individuals and 50 CRC patients), gse113486 (100 healthy individuals and 40 CRC patients), and gse124158 (275 healthy individuals and 30 CRC patients). The tissue datasets were: gse39845 (3 healthy colon tissues and 3 cancerous colon tissues), gse41655 (74 healthy colon tissues and 33 cancerous colon tissues), gse45349 (4 healthy colon tissues and 4 cancerous colon tissues), gse115513 (763 healthy colon tissues and 750 cancerous colon tissues), and gse136020 (6 healthy colon tissues and 8 cancerous colon tissues).

### 4.7. CRC Cell Lines and CSC Models

Three human CRC cell lines, HCT-116, HT-29, and T-84, provided by the American Type Culture Collection (ATCC; Manassas, VA, USA), were cultured under standard conditions in Dulbecco’s modified Eagle’s medium (DMEM) containing 10% foetal bovine serum (FBS) and 1% penicillin/streptomycin (Pen-Str P-0781; Sigma, St. Louis, MO, USA) at 37 °C under 5% CO_2_.

Secondary colonospheres were enriched in the CSCs from all cell lines with patented cell culture protocol WO2016020572A1 [90], as previously described [19]. The protocol produces colonospheres in Corning^®^ Costar^®^ Ultra-Low Attachment Six-Well Plates in DMEM/F-12 nutrient mixture without FBS, supplemented with 1× B-27 (B-27™ Supplement [50×], Minus Vitamin A; Invitrogen, Carlsbad, CA, USA), 4 ng/mL heparin (cell-culture-tested heparin sodium; Sigma), 10 µg/mL insulin (Insulin–Transferrin–Selenium [ITS-G, 100×]; Invitrogen), 1 µg/mL hydrocortisone (Sigma), 10 ng/mL epidermal growth factor (Sigma), 10 ng/mL fibroblast growth factor (Sigma), 10 ng/mL interleukin 6 (Miltenyi, Bergisch Gladbach, Germany), and 10 ng/mL hepatocellular growth factor (Miltenyi). After incubation for 72 h, trypsin (T-4049; Sigma) warmed to 37 °C was added to the culture to disaggregate the colonospheres in order to obtain secondary colonospheres.

### 4.8. RNA Extraction from Cells

Cells were disaggregated by trypsinisation, pelleted at 1500× *g* for 5 min, and washed twice in cold phosphate-buffered saline (PBS). To isolate the total RNA, 1 mL of TRI Reagent^®^ (Sigma-Aldrich) was added to the cell pellets, according to the manufacturer’s instructions. The RNA concentrations were evaluated using a NanoDrop™ 2000 spectrophotometer (Thermo Fisher Scientific).

### 4.9. Real-Time RT–PCR Assay for miRNA Expression in Cells

The RNA from HCT-116, HT-29, and T-84 cell monolayers and their respective colonospheres were diluted in nuclease-free water and resuspended at 5 ng/µL final concentration. cDNA was synthesised with the miRCURY™ LNA™ RT Kit (Qiagen), according to the manufacturer’s instructions (described previously in Section 4.4). The reactions were spiked with exogenous UniSp6 RNA (RNA Spike-In Kit, Qiagen). miRCURY LNA miRNA primers (Qiagen) were used for hsa-miR-486-5p. The U6 housekeeping gene was used for data normalisation. In the real-time RT-PCR assays, the cDNA was diluted 1:80 in nuclease-free water and 4 µL of diluted cDNA was mixed with 5 µL of PCR master mix and 1 µL of each primer pair. The thermal cycling conditions described in Section 4.4 were applied.

### 4.10. Transient Transfection with Synthetic miR-486-5p Mimics and Inhibitors

To induce the up-regulation or inhibition of miR-486-5p, an miR-486-5p mimic or inhibitor (Qiagen) were used, respectively, paired with the relative controls. The 5′-FAM-fluorescence-labelled delivery control by Qiagen was used to measure the transfection efficiency in HCT-116 monolayer cells and CSCs. Transfection was performed with TransIT-X2^®^ Transfection Reagent (Mirus Bio, Madison, WI, USA), according to the manufacturer’s instructions. The miRNA mimic and inhibitor (final concentrations of 5 nM and 50 nM, respectively) were prepared in 1.5 μL of TransIT-X2^®^ and were transfected using 100 μL of Opti-MEM medium (Gibco, New York, NY, USA) into each well of standard 24-well plates containing 6 × 10^3^ cells in 0.4 mL of medium. Before further analysis, the cells were cultured for 3 days at 37 °C under 5% CO_2_.

### 4.11. Real-Time RT–PCR Assay for Stemness Gene Expression after Treatment with miR-486-5p Mimic or Inhibitor

Total RNA was extracted from HCT-116 monolayer cells and CSCs as described in Section 4.8, before and after treatment with the miR-486-5p mimic or inhibitor, to evaluate the effect of miR-486-5p on the expression levels of stemness factors (*SOX2*, *NANOG*, *KLF4*, *OCT4*, and c*MYC*). RNA was converted into cDNA with the GoScript™ Reverse Transcription System (Promega, Madison, WI, USA), according to the manufacturer’s instructions. The cDNA (diluted 1:80), GoTaq^®^ qPCR Master Mix (Promega), and primers from the StemElite™ (Promega) *SOX2/NANOG/KLF4/OCT4/*c*MYC/GAPDH* Primer Pair (20×) were used according to the manufacturer’s instructions. *GAPDH* was used as housekeeping gene.

### 4.12. Microarray Hybridisation

The total RNA extracted from the HCT-116 monolayer cells and CSCs after treatment with the miR-486-5p mimic or inhibitor was amplified, labelled, and hybridised with the Clariom D platform (Affymetrix, Santa Clara, CA, USA), according to the manufacturer’s instructions. Briefly, a HeLa positive control and polyA RNA controls were prepared according to the manufacturer’s instructions. The Affymetrix GeneChip WT Pico Kit was used to synthesise single-stranded cDNA with the T7 promoter sequence at the 5′ end using 3 ng of total RNA for each condition (mimic, mimic control, inhibitor, and inhibitor control, in triplicate). A 3′ adaptor was added to the single-stranded cDNA, and double-stranded cDNA was synthesised with Taq DNA polymerase and adaptor-specific primers. cRNA was synthesised with T7 RNA polymerase and purified with purification beads. The cRNA was used to synthesise second-cycle single-stranded cDNA. The samples were then subjected to RNA hydrolysis with RNase H and the purification, fragmentation, and labelling of the second-cycle single-stranded cDNA. The Affymetrix GeneChip^®^ Hybridization, Wash, and Stain Kit was used to process the array. The biotin-labelled single-stranded cDNA was hybridised to the GeneChip^®^ Cartridge Array, and the array was incubated for 16 h in an Affymetrix GeneChip^®^ Hybridization Oven 645 at 45 °C with rotation at 60 rpm. The array was scanned with the Affymetrix Expression Console™ software (v.1.4; Affymetrix London, UK). The working hypothesis that miR-486-5p affects one or more of the principal genes involved in one of the four principal pathways of the CSC phenotype (Wnt, Notch, Hedgehog, and TGF-β), with the same trends in the monolayer cells and CSCs, was then tested.

### 4.13. Statistical Analysis

To evaluate the whole serum miRNome differential expression in sera of CRC patients under different conditions and in the controls, the obtained sequences were mapped onto the reference human genome hg38 version (accession GCA_000001405.15) [91], and the raw data were normalised with the trimmed mean of M values (TMM) method [92]. The fold change (FC) values were compared with Student’s independent *t* test and analysis of variance (ANOVA). A two-sided *p* value < 0.05 was considered statistically significant, and the false discovery rate (FDR) [93] was calculated to account for multiple comparisons.

Real-time RT–PCR was used to evaluate the differential expression of miR-486-5p in the sera of the different groups of patients and controls (Section 4.4), and its differential expression in the three cell models under the two sets of conditions examined (Section 4.10). The same technique was also used to test the differential expression of stemness factors in HCT-116 cells (monolayers and CSCs) after treatment with the miR-486-5p mimic or inhibitor (Section 4.11). The real-time RT–PCR assays were run in triplicate, and the mean cycle threshold (Ct) value for each miRNA under each set of conditions was used to determine the FC with the 2^−ΔΔCt^ method. The FCs were compared with Student’s unpaired *t* test. A two-sided *p* value < 0.05 was considered statistically significant.

To extend the population of study, a random-effects meta-analysis (Section 4.5) was performed by computing the log FC of miR-486-5p in CRC patients vs. controls.

To evaluate the miRNome in stool samples (Section 4.6), small-RNA-Seq pipeline analyses were performed with a previously described approach [86]. The differential expression analysis was performed with the DESeq2 R package v.1.22.2 [94] using the likelihood ratio test (LRT) function, adjusting for age and sex as covariates. A gene was defined as differentially expressed if it was associated with a Benjamini–Hochberg (BH)-adjusted *p* value < 0.05 and was supported by at least a median number of reads > 20 within at least one of the sample groups considered. Data from microarray (Section 4.13) were normalised with the Affymetrix Expression Console™ software (v.1.4; Affymetrix UK Ltd.), which provides signal estimation and quality control functionalities for the GeneChip Expression Arrays, and the Affymetrix Transcriptome Analysis Console (TAC) Software version 4.0.2 (Thermo Fisher). We performed a genome-wide scan to identify those genes with expression patterns consistent with the working hypothesis derived from the genomic analysis, the meta-analysis, and from functional analysis of HCT-116. To this end, after the treatments with mimics or inhibitor of miR-486-5p, we identified genes that were progressively inhibited or promoted by miR-486-5p, both in monolayer and CSC, by using the Jonckheere–Terpstra test for trend, with the empirical *p* value determined after 10,000 permutations. Subsequently, we investigated whether the selected genes were enriched in the four main pathways of the CSCs phenotype (Wnt, Hedgehog, Notch and TGF-β), using the weighted Kolmogorov–Smirnov enrichment test [95], considering enrichment *p*-value < 0.05 as statically significant.

## 5. Conclusions

We have investigated the expression pattern of miR-486-5p in CRC on three different matrices, serum, stool, and tissue, finding in CRC patients an up-regulation of this miRNA in the first two matrices and a down-regulation in the third, in respect to controls. Then, we examined miR-486-5p role in the occurrence of the CSC phenotype in CRC cells, through in vitro mechanistic and genomic studies. We also observed a marked down-regulation of miR-486-5p in the sera of patients with metastatic cancer, suggesting the involvement of miR-486-5p in the inhibition of metastasis. This hypothesis is supported by our genomic data, which indicated a role for miR-486-5p in the biology of CSCs by negatively regulating the expression of stemness factors and the main pathways of CSCs: Wnt, Notch, Hedgehog, and TGF-β. In conclusion, our preliminary data could provide a step forward in the understanding of the biomolecular mechanisms of miR-486-5p in CRC and to suggest further investigation to evaluate it as possible prognostic and diagnostic biomarker.

## 6. Strengths and Limitations of the Study

The present study aiming to improve the current knowledge about the role of miR-486-5p in CRC and in the CSC presents some limitations: this study sample size was enough to identify differential expression comparing early stage patients with healthy controls as well as comparing advanced with non-metastatic tumors. Although further studies, with larger sample size, are needed to clarify miR-486-5p role in CRC diagnosis and prognosis, our results suggest its usefulness in the clinical practice, possibly in combination with other well-known biomarkers (for example CEA). Moreover, in this study, we were not able to measure miR-486-5p in serum, tissue and stool samples from the same patients. Concerning the in vitro part of this study, although our results confirm data from the literature in the field, experiments were performed on a selected cell line, specifically HCT-116, introducing the need to further confirm our observations about cell migration and tumor sphere formation in different models.

## Figures and Tables

**Figure 1 cancers-12-03432-f001:**
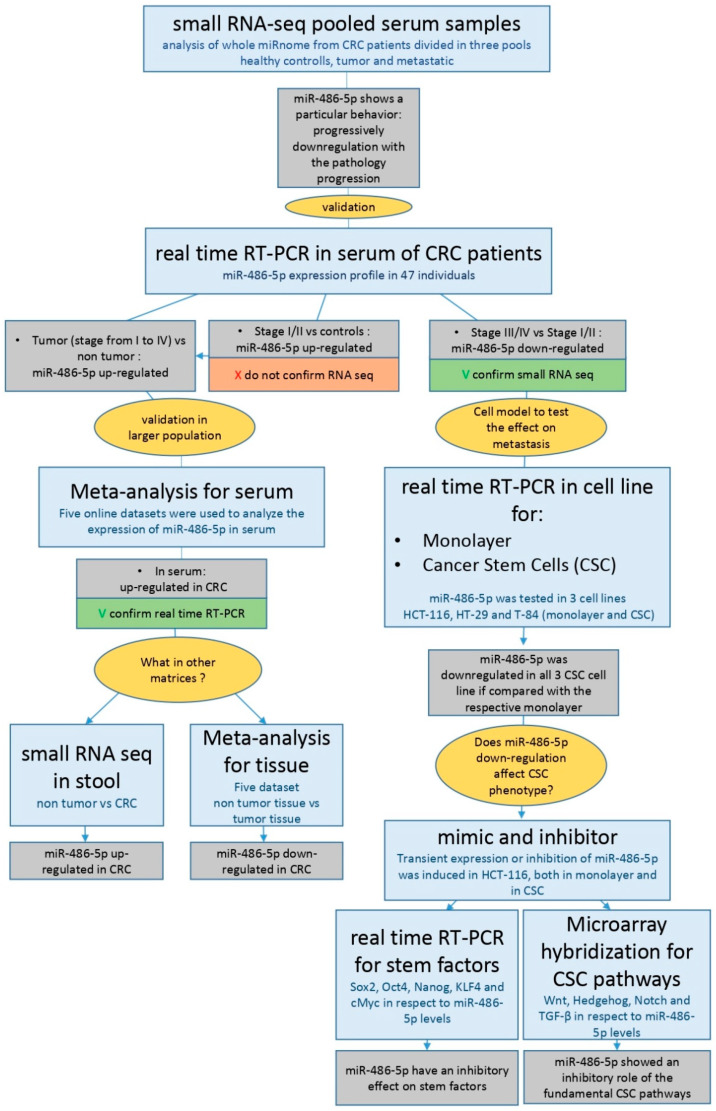
Workflow with all steps and results of the present study.

**Figure 2 cancers-12-03432-f002:**
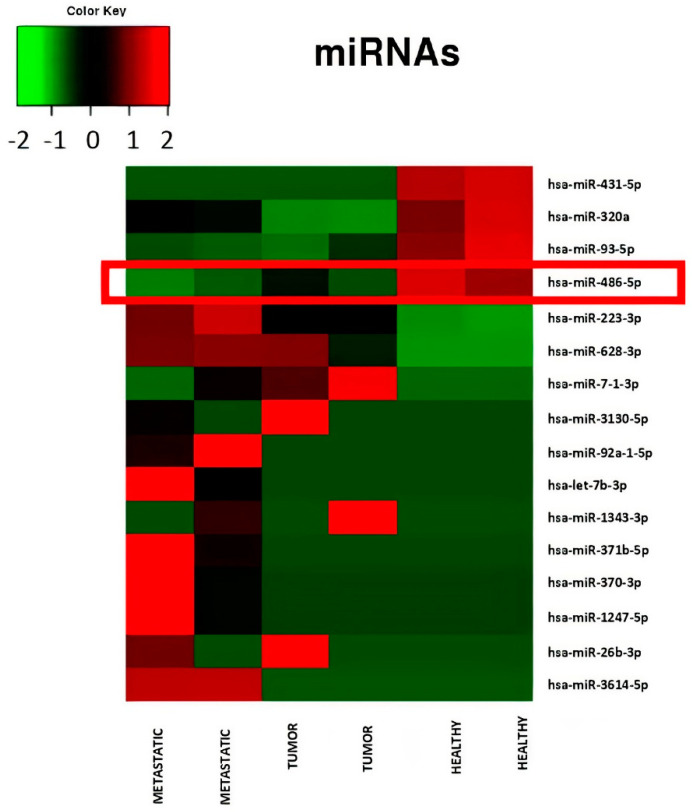
Heatmap relative to miRNA expression levels measured by miR-Seq and significantly altered in serum pools from patients in different condition: controls, patients with low stage CRC (stage I to II), patients with advanced stage CRC (stage III to IV). A progressive down-regulation of miR-486-5p from controls to advanced stage CRC patients is showed.

**Figure 3 cancers-12-03432-f003:**
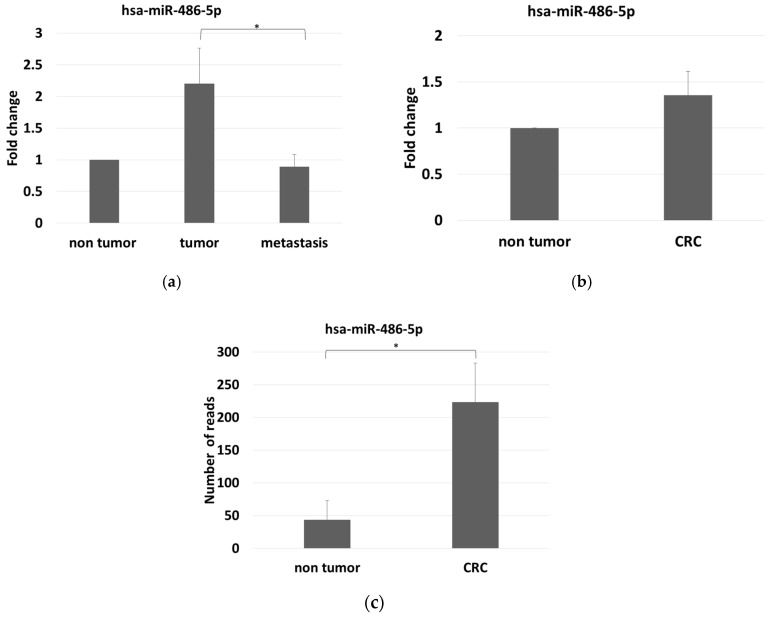
(**a**) miR-486-5p expression levels among the three groups analyzed, measured by Real-time RT-PCR in serum, and normalized with the mean expression levels of hsa-miR-342-3p, hsa-miR-223-3p and hsa-miR-93-5p; (**b**) miR-486-5p expression levels by comparing control and tumor group, as measured by Real-time RT-PCR in serum and normalized with the mean expression levels of hsa-miR-342-3p, hsa-miR-223-3p and hsa-miR-93-5p; (**c**) stool miR-486-5p expression levels as measured by small RNA-sequencing, in controls and CRC patients. The graphic reports average values with standard errors. The symbol * indicates differences supported by statistical significance.

**Figure 4 cancers-12-03432-f004:**
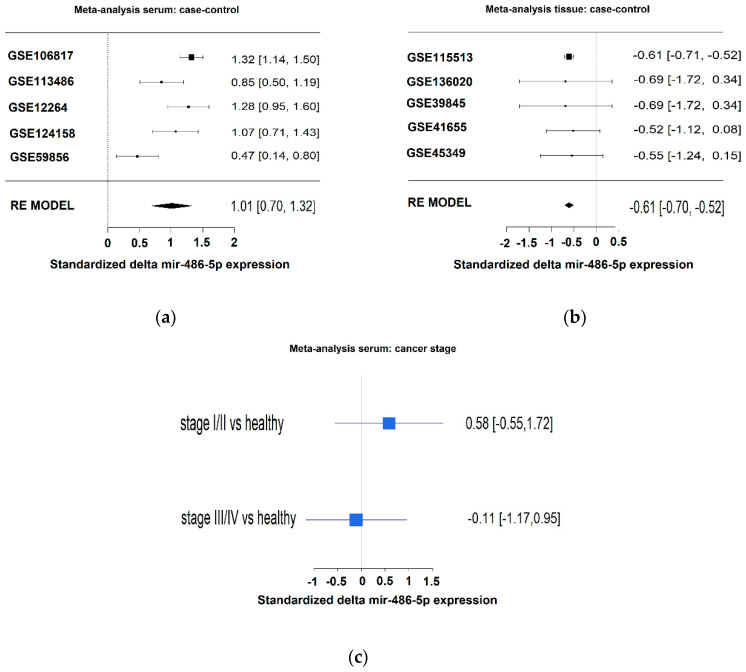
(**a**) Meta-analysis of miR-486-5p levels in serum by the use of five datasets. Higher levels of miR-486-5p were observed in the case groups; (**b**) meta-analysis of miR-486-5p levels in tissues by the use of five datasets. Lower levels of miR-486-5p were observed in the case group; (**c**) expression levels of miR-486-5p related to stage in GSE39845 dataset from the GEO Database. The patients were divided in two groups: low stage (stage I and II) and advanced stage (stage III and IV).

**Figure 5 cancers-12-03432-f005:**
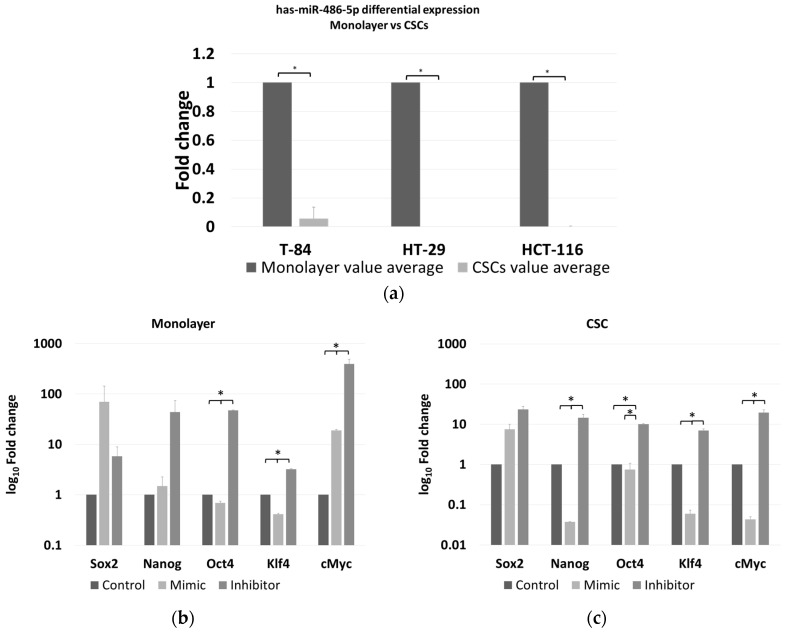
(**a**) Differential expression levels of miR-486-5p between monolayer and CSC derived from T-84, HT-29, HCT-116 cell lines as measured by Real-time RT-PCR. A down-regulation of miR-486-5p in all CSC models is evident in comparison to the respective monolayer; Real-time RT-PCR quantification of stemness factors (Sox2, Nanog, Oct4, Klf4 and cMyc) in monolayer (**b**) and in CSCs (**c**) after treatment with miR-486-5p mimic or inhibitor; each treatment was normalized to the respective control (mimic vs. control mimic and inhibitor vs. control inhibitor) represented in this figure as unique control (both controls were indicated as reference and imposed equal to 1). (**d**) real-time RT-PCR quantification and comparison of the differential expression of stemness factors (Sox2, Nanog, Oct4, Klf4 and cMyc) in monolayer and CSCs after control mimic treatment; (**e**) real-time RT-PCR quantification and comparison of the differential expression of stemness factors (Sox2, Nanog, Oct4, Klf4 and cMyc) in monolayer and CSC after control inhibitor treatment. The symbol * indicates differences supported by statistical significance.

**Figure 6 cancers-12-03432-f006:**
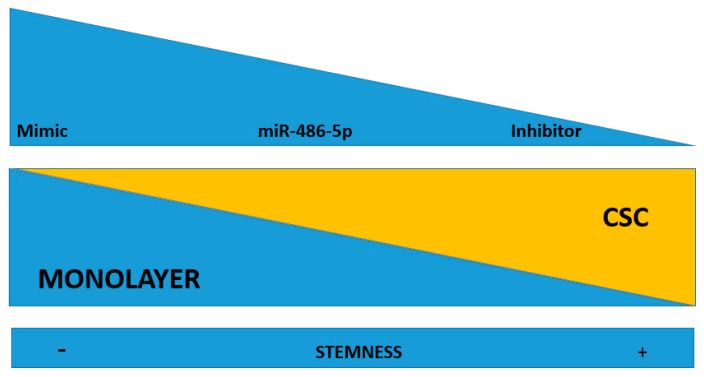
Working hypothesis: miR-486-5p may play an inhibitory effect on the stemness of CSCs. When it is up-regulated by mimic, there is an induction of stemness loss. When the miRNA is down-regulated (by an inhibitor), it induces stemness in the recipient cells. miR-486-5p inhibition drives to CSC phenotype, while its promotion inhibits the CSC phenotype.

**Figure 7 cancers-12-03432-f007:**
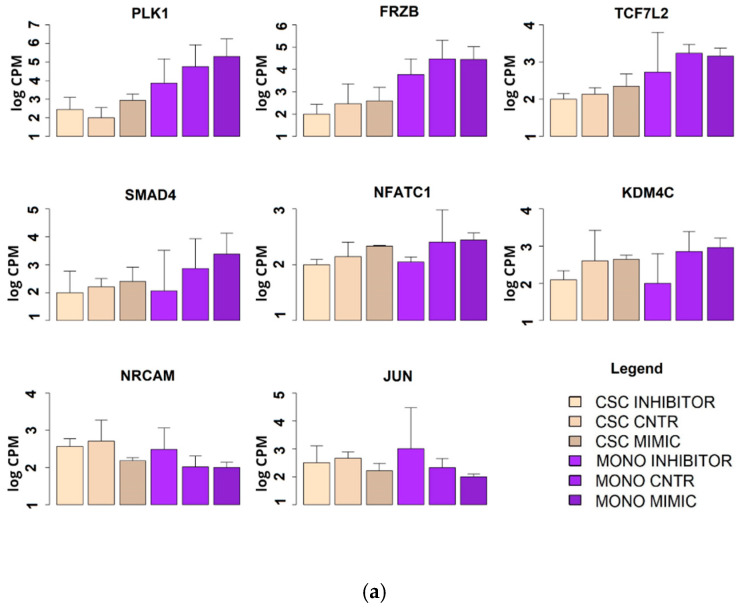
Expression levels of the genes involved in the four main regulatory pathways of the CSC phenotype (Wnt, Hedgehog, Notch and TGFβ) as measured by microarray. Expression levels were measured in two models of HCT-116: CSC and monolayer after treatment with has-miR-486-5p mimic or inhibitor. (**a**) Statistically significant genes with a progressive increment or decrement in the passage from monolayer to CSCs; (**b**) statistically significant genes with the same behavior between monolayer and CSCs.

**Figure 8 cancers-12-03432-f008:**
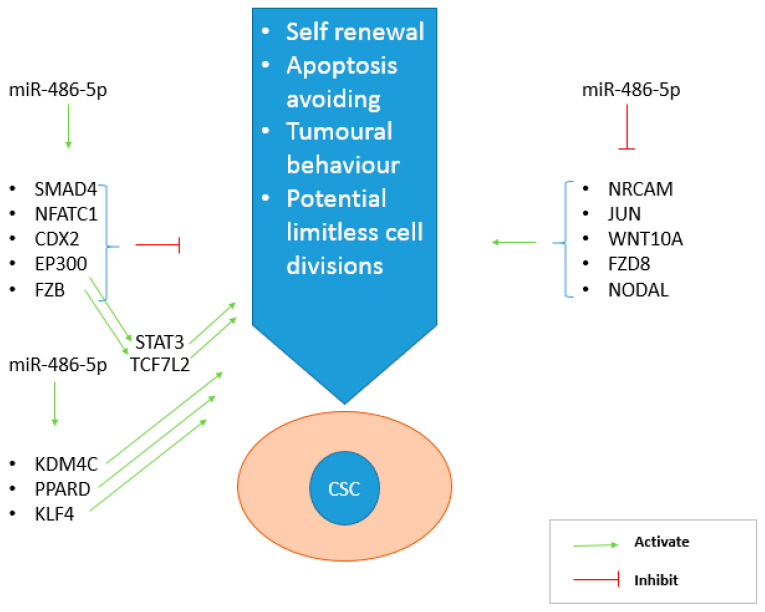
Main genes regulated by miR-486-5p thanks to its inhibitory role on CSC phenotype. On the left, the genes activated by the miRNA, on the right those inhibited. miR-486-5p promotes the effect of CSC inhibiting genes (*SMAD4*, *NFATC1*, *CDX2*, *FZB* and *EP300*), but also inhibits the action of CSC promoting genes (*NRCAM*, *JUN*, *WNT10A*, *FZD8*, *NODAL*).

**Table 1 cancers-12-03432-t001:** miR-Seq: average number of copies, *p* and adjusted *p* of most significantly dysregulated miRNAs.

miRNA	*p* Value	*p* Adj	Metastatic_1	Metastatic_2	Tumor_1	Tumor_2	Healthy_1	Healthy_2
hsa-miR-3614-5p	7.57797 × 10^−9^	2.50831 × 10^−6^	588.15	602.64	0	0	0	0
hsa-miR-1247-5p	4.44313 × 10^−8^	3.91482 × 10^−6^	923.17	124.99	0	0	0	0
hsa-miR-26b-3p	2.49849 × 10^−8^	3.91482 × 10^−6^	405.75	0	693.60	0	0	0
hsa-miR-370-3p	4.7309 × 10^−8^	3.91482 × 10^−6^	930.61	138.38	0	0	0	0
hsa-miR-371b-5p	3.02639 × 10^−7^	2.00347 × 10^−5^	655.15	196.42	0	0	0	0
hsa-miR-92a-1-5p	1.18488 × 10^−6^	6.53656 × 10^−5^	201.01	620.50	0	0	0	0
hsa-miR-1343-3p	1.58942 × 10^−6^	7.51568 × 10^−5^	0	178.56	0	449.91	0	0
hsa-let-7b-3p	1.87635 × 10^−6^	7.76339 × 10^−5^	450.42	111.60	0	0	0	0
hsa-miR-3130-5p	3.90736 × 10^−6^	0.000143704	171.23	0	610.03	0	0	0
hsa-miR-93-5p	3.5912 × 10^−5^	0.001188687	23,101.53	21,016.67	18,685.45	26,506.86	47,283.15	57,832.90
hsa-miR-223-3p	0.000104139	0.003133635	355,620.64	417,538.85	288,671.90	294,471.36	175,026.35	162,243.02
hsa-miR-431-5p	0.000641093	0.017683487	0	0	0	0	660.62	748.75
hsa-miR-320a	0.00126751	0.032272748	129,310.50	120,542.28	57,794.55	53,603.04	191,473.93	241,138.63
hsa-miR-486-5p	0.001484273	0.03407904	708,832.85	796,133.48	937,832.71	815,698.98	1,422,293.08	1,307,148.95
hsa-miR-628-3p	0.001544367	0.03407904	416.91	437.47	434.54	183.78	0	0
hsa-miR-7-1-3p	0.002010025	0.041582387	0	441.94	626.75	1184.98	0	0

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
