# Peer review of "The Inhibitory Role of miR-486-5p on CSC Phenotype Has Diagnostic and Prognostic Potential in Colorectal Cancer"

_cancers, 2020, doi:10.3390/cancers12113432_

Round 1
Reviewer 1 Report
The manuscript entitled “CSC phenotype and Metastasis in Colorectal Cancer: the inhibitory role of miR-486-5p” by Andrea Pisano et al.” addresses an important topic in colorectal cancer research, by studying the role of this miRNA in the CSC phenotype, for the discovery of new diagnostic and prognostic biomarkers and provide interesting results. However, some critical points should be improved:
Major concerns:
- The authors used the term metastatic, or high grade referring to stage III-IV patients. This could lead to confusion with the grade staging of tumors that it is actually based on cell differentiation. https://www.cancer.net/cancer-types/colorectal-cancer/stages. Moreover, according to the TNM classification, stage III patients are not considered as metastatic patients. If the authors want to include stage III and IV together, they should should terms such as advanced CRC or advanced stage CRC.
- In the miRNA seq from serum, miR-486 was upregulated in healthy patients vs CRC patients, showing lower levels in the metastatic group. However, in the RT-PCR from the same material, this miRNA was upregulated in CRC patients vs controls and in early stages in comparison with advanced stages. The authors report this contradiction but did not discuss the potential explanation for why miRNA seq and RT-PCR showed opposite results. This needs to be addressed.
- As this study “aimed to identify miRNA role to improve the diagnostic and/or prognostic tools in CRC”. The authors should provide results that support this aim. When analyzing the prognostic factor of this miRNA. The authors did not show any analysis of the relation between the expression of this miRNA and the outcome, development of recurrence or progression, patient survival, etc. They only provided a statistical difference between stage I-II and stage III-IV. For this reason, to make this kind of statement they should provide, at least, a univariate analysis of the progression-free survival, recurrence-free survival, or overall survival. The comparison with prognostic biomarkers currently used in the clinical management of CRC patients, such as the carcinoembryonic antigen (CEA), would highly improve the quality of this research article.
- In the same way, the authors could easily report the sensitivity and specificity of the serum miR-486 at the diagnosis of CRC in comparison with healthy controls based on their RT-PCR results. This would increase the potential future application of their results. Indeed, the comparison with current clinical used biomarkers would be also an additional interesting point.
- The authors describe the importance of miR-486-5p in the biology of CSC, however they did not provide any further characterization of how this miRNA alter the metastatic potential of these CSC. They only showed the alteration of the genes involved in the CSC pathway. Importantly, the demonstration of the role of this miRNA in the acquisition of a more malignant phenotype associated to metastatic potential could be demonstrated through experiments such as migration, proliferation, invasion or specifically tumorspheres formation, as previously described (Serrano-Oviedo L. Identification of a stemness-related gene panel associated with BET inhibition in triple negative breast cancer. 2020 Cellular Oncology volume 43, pages431–444; Bono B, Cells with stemness features are generated from in vitro transformed human fibroblasts Scientific Reports volume 8, Article number: 13838 (2018); Alowaidi, F, Assessing stemness and proliferation properties of the newly established colon cancer ‘stem’ cell line, CSC480 and novel approaches to identify dormant cancer cells Oncology Reports April 23, 2018. In case these experiments are not performed, the statement about the role in the metastasis should be downplayed.
Minor concerns:
- Despite including a total of 92 references, some parts of the introduction need a reference such as line 84-87. I believe that the introduction and references provided could be improved and be more specific about CRC. For example, the authors reported 2 articles showing evidence that miRNAs are good candidate markers, one for diagnosis and one for prognosis, but one was related to lung cancer indeed, that article analyzed also miRNA 486-5p which had no impact on the final model for diagnosis. There are more specific reports on CRC showing the diagnostic and prognostic value of miRNAs in CRC such as H. Imaoka et al. (2016) doi.org/10.1093/annonc/mdw279, Jane V. Carter et al. (2017) doi: 10.1097/SLA.0000000000001873, or Diego de Miguel Pérez et al. (2020) doi: 10.1038/s41598-020-60212-1.
- The authors performed a serum miRNA-seq from pooled serum samples. In Figure 2, they show a heat-map of 16 miRNAs but they did not report any supplementary table to indicate if those miRNAS showed any differential expression between groups. Was miRNA-486-5p the only differently expressed between the 3 groups? In the figure caption they say those miRNAs are significantly altered, could they provide the rationale for the representation of those 16 miRNAs, were they underexpressed or overexpressed? This could be better understood if the figure was cleared as the color key is unreadable. Red means lower expression or higher?
- Were the patients and samples collected prospectively or retrospectively? This should be included to understand potential study bias.
- Line 129, the authors used miR-342 as reference and described than tumors (tumor and metastatic has differential expression than controls but p value >0.05, so this means no difference. Is this a typographical error?
- Line 177: Describing every fold change during the results when it’s also available in the figures increase the complexity of the text and complicates the process of understanding the results. Avoid duplicates in the text and figures. In the same way, it feels repetitive to divide table 1 into 2, when they actually only showed the p-values from figure 5. They may introduce these tables as supplementary material.
- Line 219: miR-486-5p inhibits the expression of 7 genes (NRCAM, JUN, WNT10A, FZD8, and NODAL). Which one are the other 2 genes missing? Same in Line 355: 4 genes why there are 5?
- Line 186: “except for SOX2, which was most strongly expressed in the mimic-treated cells”. Despite observing increased levels in the graph, this difference is not statistically supported. No statistical differences were found between the mimic, control nor inhibitor for SOX2.
- Figure 5 a and b showed the log10 fold change of gene expression in the graph but also in a small table under it, which is in pretty bad quality and values cannot be read. Duplicate values in the graph and the table are not necessary.
- Line 261: The authors reported “when the CRC group were merged in a single group (all stages) and compared with the control group, miR-486-5p was upregulated in the control group”. This is contradictory to what was described before. As observed in figure 3c, miR486 is upregulated in the CRC group vs controls.
- Figure 3 seems repetitive. Why the authors report different analysis based on different miRNAs used as endogenous? The authors should select the best and stick to them. Morever, which miRNA or miRNAs were used in the in vitro experiments? Caption in figure 3e does not report that was analyzed in stool. This should be included in the caption and in the figure to clarify the origin of the miRNA.
- In line 262, the authors report the number of patients analyzed in the meta-analysis for the first time, this number of patients analyzed should be also included in the methods.
- In the meta-analysis, the authors describe that serum miR-486 has a risk ratio of 1.01 and that is overexpressed vs controls. However, the value observed in Figure 4A is 0.87. Moreover, why dataset GSE39833 was not included in the analysis in Figure 4A and 4B? Does this metaanalysis include only circulating free miRNAs or also exosomal miRNAs? This should be specified as it may help explain why tissues and serum levels does not correlate.
- The authors focus on the role of this miRNA at modulating CSC phenotype and metastasis. Was the presence of CSCs analyzed in these patients? Does the miRNA expression correlate with the presence of those cells?
- In section 2.6. The authors only focus on the HCT-116 cell line, stating that the selection was based on previous results, however, T-84 and HT-29 also showed clear downregulation of miR-486 in the CSC model. If they are going to only analyze one of the cell lines, they should justify it.
- Quality of figures must be improved. Especially figure 5, text is too small, cannot be read. Moreover, p-value bars in Figure 5c are badly located and do not match the gene expression bars.
Reviewer 2 Report
The authors first investigate the miRNAome in 47 samples from serum samples (10 non tumor; 13 stages I-II; 24 stages III-IV). In parallel they study expression of 137 stools, 79 from healthy subjects and 58 from CRC but they do not report if there is any difference according tumour stage. It would be more consistent to study tumour, serum and stools from the same patients. However, I understand that it would be very difficult for the authors to guide the manuscript in this direction
They focus in the study of miR-486-5p expression. As the authors state this miRNA has been previously studied in different tumours with results similar to those reported in this manuscript. Thus, there are not new results. They describe that miR-486-5p is upregulated in serum and stool samples of CRC patients, but expression decreases as the disease progressed. The authors confirm this observation by meta-analysis. They state that the downregulation of miR-486-5p in sera of metastatic cancers implies the involvement of this miRNA in metastasis. The problem is that the authors are unable to define the origin of the miRNA present in serum and stools. It could be an epiphenomenom related to cells other than cancer cells. They do not perform any analysis to corroborate the origin of this circulating miRNA. I consider that they should be more cautious in the discussion.
Finally, they study miR-486-5p expression in three CRC cell lines and CSCs from HTC116 and show that this miRNA regulates genes expressed in CSCs. The authors do not state the other pathways that are regulated by this miRNA. I consider that the observation is very weak to elaborate a hypothesis as they only study one cell line and results could not be representative of what happens in vivo. As a matter of fact, the results could be different if analyzed different CRC derived cell lines.
In summary, this is a manuscript with a good methodology, that report some new information related to miR-486-5p, but there are strong doubts related to its possible role as specific biomarker of CRC.
Reviewer 3 Report
The authors of current paper had investigated the possible role of miR-486-5p in diagnosis and cancer stemness cell (CSC) phenotype and metastasis of colorectal cancer (CRC). The paper is well organized, and logically presented.
Some concerns about the data:
- the quality of Fig.2,5, and 7 need to be improved as it is difficult to see the data presented in their current forms.
- In Fig.3, it was shown that has-miR-486-5p level was higher in the sera from tumor patients than from the healthy controls and from patients with metastasis. How about the levels of has-miR-486-5p in healthy control vs those from metastasis? It looked like that the healthy control had similar level of has-miR-486-5p to the level detected in the sera of patients with metastasis. If so, how is it to be differentiated between control and metastasis? The statistical markers are missing in C and D.
- In the results (Fig.7) from the microarray genomic analysis, it is unclear how the effect of miR-486-5p on those genes described in the stemness pathways was related to the changes of those genes in monolayer and/or cancer stem cells (CSC). The results need to be explained a clearer way to show what was changed and how those changes were related to miR-486-5p.
Author Response
The authors thank the reviewers for the careful revision of our manuscript which allowed to improve the quality of our manuscript.
Below, our point-by-point response to their comments. We indicate the lines and the section of the change operate on the manuscript, the line numeration is referred to the text with "Track Changes" of Microsoft Word active.
Comments and Suggestions for Authors
The authors of current paper had investigated the possible role of miR-486-5p in diagnosis and cancer stemness cell (CSC) phenotype and metastasis of colorectal cancer (CRC). The paper is well organized, and logically presented.
Some concerns about the data:
- the quality of Fig.2,5, and 7 need to be improved as it is difficult to see the data presented in their current forms.
We totally agree with your remarks, and provided to improve the quality and legibility.
- In Fig.3, it was shown that has-miR-486-5p level was higher in the sera from tumor patients than from the healthy controls and from patients with metastasis. How about the levels of has-miR-486-5p in healthy control vs those from metastasis? It looked like that the healthy control had similar level of has-miR-486-5p to the level detected in the sera of patients with metastasis. If so, how is it to be differentiated between control and metastasis? The statistical markers are missing in C and D.
We thank the reviewer for his comment. In the comparison of healthy vs. metastatic patients we did not observed statistically significant differences. In the main text we provided a possible biological interpretation of these results (lines 358-369), as well as a discussion about the diagnostic and prognostic role of this biomarker (302-315), which is likely able to discern healthy vs. early stage patients, and early stage patients vs. metastatic. The described trend is in line with our literature based interpretation of the phenomena
- In the results (Fig.7) from the microarray genomic analysis, it is unclear how the effect of miR-486-5p on those genes described in the stemness pathways was related to the changes of those genes in monolayer and/or cancer stem cells (CSC). The results need to be explained a clearer way to show what was changed and how those changes were related to miR-486-5p.
We thank the reviewer for his comment, Figure 7 refers to an analysis in which we compared gene expression of a selected set of genes pertaining to CSC-related pathways in different cell lines in which
we induced both up regulation (by mimic) and down regulation (by inhibitor) (section 4.10) of mir486-5p. We reported in figure 7 the results of these analyses in both monolayers and CSCs models. In figure 7A we displayed genes that were progressively down or up regulated from the presence of miR-486-5p in the passage from monolayer to CSC, while in figure 7B we displayed genes with similar behavior in both the models: monolayer and CSC. We modified the main text to explain clearly this concept (line 235-238 and 245-246 in paragraph 2.7 in results).

Round 2
Reviewer 1 Report
The authors have addressed the critical points that needed to be improved.
Author Response
We thanks the Reviewer.
Reviewer 3 Report
Please cross one "suffered" from line 217.
Author Response
We thank the Reviewer, and we make the correction you suggested to us ( see line 229).